# A Comparative Effect of 12-Week Dietary Intervention of Policosanol (Raydel^®^) and Red Yeast Rice (RYR, Kobayashi) in Managing Dyslipidemia and Organ Damage in Hyperlipidemic Zebrafish

**DOI:** 10.3390/ph18020200

**Published:** 2025-02-01

**Authors:** Kyung-Hyun Cho, Ashutosh Bahuguna, Ji-Eun Kim, Sang Hyuk Lee, Yunki Lee, Cheolmin Jeon

**Affiliations:** Raydel Research Institute, Medical Innovation Complex, Daegu 41061, Republic of Korea

**Keywords:** policosanol, red yeast rice, liver, kidney, brain, oxidative stress, paraoxonase, dyslipidemia, reproductive organs

## Abstract

Background: A comparative 12-week dietary intervention of red yeast rice (RYR, *Beni-koji*, Kobayashi, Japan) and Cuban policosanol (PCO, Raydel^®^, Thornleigh, Australia) was assessed for dyslipidemia, antioxidant status, and vital organ functionality in hyperlipidemic zebrafish. Methods: Hyperlipidemic zebrafish were supplemented with a high-cholesterol diet (HC, final 4%, *w*/*w*) infused with either a powdered RYR tablet (final 1.0%, *w*/*w*), a PCO tablet (final 1.0%, *w*/*w*), or a combination of 0.5% (*w*/*w*) each of RYR and PCO powder for 12 weeks. Subsequently, blood and organs were collected and processed for biochemical and histological examination. Results: RYR and PCO consumption showed a substantial effect against HC-induced hyperlipidemia by reducing the total cholesterol (TC), triglyceride (TG), and low-density lipoprotein cholesterol (LDL-C). Exclusively, PCO supplementation revealed a significant elevation in the HC-diminished high-density lipoprotein cholesterol (HDL-C). In addition, PCO supplementation showed a significant elevation in plasma ferric ion-reducing ability (FRA) and sulfhydryl content, as well as alleviating the blood glucose level of hyperlipidemic zebrafish. The most noteworthy impact, with a significant two-fold (*p* < 0.001) augmentation of HC-diminished plasma paraoxonase (PON) activity, was observed in response to PCO. In contrast, the RYR supplementation failed to establish curative effects against HC-disturbed plasma antioxidant variables and blood glucose levels. The histological outcome revealed a severe toxicological impact of the RYR on the liver, reflected by fatty liver changes and three-fold heightened IL-6 production compared to HC control. Contrastingly, PCO exhibited significant hepatoprotection and effectively neutralized the hepatic toxicity triggered by HC and RYR. Also, RYR showed kidney atrophy, intense ROS generation, apoptosis, and senescence. Conversely, the PCO supplementation protected the kidney from HC- and RYR-induced toxicity. Likewise, PCO supplementation notably alleviated histological alterations and oxidative stress in the brain, ovary, and testis of hyperlipidemic zebrafish. Conclusions: This comparative study establishes PCO’s therapeutic effect against the challenges posed by HC, while RYR emerged with serious toxicological concerns towards the liver, kidney, and other organs of hyperlipidemic zebrafish.

## 1. Introduction

Dyslipidemia is commonly defined by increased levels of total cholesterol (TC), triglyceride (TG), and low-density lipoprotein cholesterol (LDL-C) and decreased levels of high-density lipoprotein cholesterol (HDL-C). Dyslipidemia is associated with the pathogenesis of many diseases, and its main impact is observed at the onset of coronary heart disease and atherosclerosis [1,2]. Statins, either alone or in combination with other medications, are commonly prescribed for the management of dyslipidemia [3,4]. However, their long-term use has been associated with potential adverse effects on liver function and muscle health with new-onset diabetes and worsening glycaemia [3,5,6,7]. As safe lipid-lowering agents, various nutraceutical and functional foods like policosanol (PCO), red phytosterols, seaweed, rice bran oil, hawthorn fruit, red yeast rice (RYR), and garlic have been reported [8].

Among them, PCO has been extensively explored in preclinical and clinical studies [9,10,11]. PCO is a mixture of long-chain aliphatic alcohols (LCAAs) extracted from a variety of sources [10,12], and its composition substantially varies based on the source material’s geographical location, environmental conditions, and method of extraction [11,13]. For instance, Cuban PCO extracted from sugar cane wax is a typical mixture of eight LCAAs (C24–C34) that substantially differs from the PCO obtained from the sugar cane of China and has distinct functionality [14]. PCO is well known for its diverse functionality, including the inhibition of the platelet aggregation effect, the improvement of neurological function in patients with ischemic stroke, and impacts on Parkinson’s and Alzheimer’s disease [11]. Also, PCO’s impact on controlling blood pressure through the modulation of aldosterone [15] has been well described in clinical trials with healthy Korean [15] and Japanese participants [16].

Nevertheless, the most impactful role of PCO is documented for dyslipidemia [10]. Interestingly, PCO elevates the blood HDL-C level and impacts HDL-C-coupled antioxidant paraoxonase (PON)-1 [17]. In addition, PCO down-regulates the expression of apolipoprotein B (apo-B) and enhances the expression of LDL receptors, facilitating LDL-C uptake by the liver [12]. Moreover, PCO impacts hepatic cholesterol catabolism, rendering it into bile acid and promoting its excretion through the feces [12,18].

The safety aspect of PCO is studied in a wide range of animal models like rats, dogs, and monkeys, which consumed 1500 times more than the usual human dose and showed no adverse events [19]. Also, no adverse effect was observed concerning reproductive health or teratogenic and developmental defects [20,21,22]. Similarly, in human clinical trials, PCO emerged as a safe supplement [19]. In a post-marketing study with 27,879 patients, only 0.07% reported weight loss and polyuria, while 0.05% reported polyphagia [23]. Owing to several beneficial functions and nontoxic effects, PCO is gaining acceptability as a nutraceutical globally, as evidenced by the approval of PCO by 25 countries as a safe cholesterol-lowering agent [24].

Like PCO, red yeast rice (RYR) has been reported as a nutraceutical/functional food with blood cholesterol-lowering effects [25]. RYR is produced through the fermentation of rice by mold (*Monoscus purpureus*) [26] and marketed under different brand names [8] like “*Beni-koji* Coleste help” (developed and marketed by Kobayashi Co., Ltd., Osaka, Japan) [27]. RYR contains sterols, isoflavones, fatty acids, and monacolin K, with a claimed lipid-lowering effect [25]. However, the most noteworthy effect was noticed for monacolin K (analogous structure to lovastatin) [25]. Studies documented RYR’s broad pharmacological role against dyslipidemia, fatigue, osteoporosis, and hypertension [28]. Nevertheless, hepatotoxicity, respiratory injury, reproductive toxicity, and musculoskeletal and gastrointestinal issues highlight concerns about RYR consumption [29]. Given the possible toxicity and health implications, the United States Food and Drug Administration (FDA) has issued specific guidelines against RYR supplementation for dyslipidemia [30].

Although various studies (non-clinical and clinical) have been conducted using PCO or RYR, a direct comparative evaluation between the two and on the effect of their combination (RYR + PCO) has not yet been performed. In light of this, the present study aimed to assess the efficacy of RYR and PCO, individually and in combination (RYR + PCO), in addressing high-cholesterol-diet-induced dyslipidemia and altered antioxidant parameters in adult zebrafish. Additionally, the comprehensive effect of RYR, PCO, and RYR + PCO supplementation on the functionality and histological changes in the liver, kidney, brain, and reproductive organs of hyperlipidemic zebrafish was assessed.

## 2. Results

### 2.1. Zebrafish Mortality and Body Weight

As depicted in Figure 1A, HC consumption adversely affects zebrafish survivability, which starts to decline after 4 weeks (89.3%), further declines to 78.6% at 10 weeks, and thereafter remains constant until the final 12 weeks of feeding. The supplementation of RYR with HC severely impacted zebrafish survivability, where a linear decline in survivability was noticed from week 4, reaching 67.8.% survivability at 12 weeks. In contrast, PCO supplementation effectively improved zebrafish survivability, which was ~10% and 26% higher than the survivability observed in the HC and HC + RYR groups at 12 weeks. Similarly, PCO, in combination with RYR, also improved survivability, which was 21% higher than the survivability noticed in the HC + RYR group.

As compared to the HC group, a non-significant (*p* > 0.05) body weight change was noticed among the different groups following 12 weeks of supplementation of the respective diets (Figure 1B). However, compared with the 1.0% RYR-supplemented group, a significant 16% (*p* < 0.05) reduced body weight was noticed in the PCO group.

### 2.2. Plasma Lipid Profile

The HC diet elevated the blood TC level (528.9 mg/dL), which was significantly reduced by 8.4%, 17.9%, and 12.9% by the dietary supplementation of RYR, PCO, and RYR + PCO, respectively (Figure 2A). Likewise, RYR, PCO, and RYR + PCO consumption significantly (*p* < 0.001) reduced the HC-induced TG level (503.9 mg/dL) by 13.9% (433.6 mg/dL), 24.5% (380.2 mg/dL), and 16.9% (418.6 mg/dL), respectively (Figure 2B).

In addition, HC significantly diminished HDL-C levels by 4-fold (*p* < 0.001) and 3.5-fold (*p* < 0.001), augmented by the consumption of the PCO and RYR + PCO, respectively (Figure 2C). Contrary to this, a non-significant 1.2-fold (*p* > 0.05) elevated HDL-C level was observed in the RYR group compared to the HC group.

The LDL-C level was notably high in the HC (266.4 mg/dL) group, which was significantly (*p* < 0.001) diminished by 1.5-fold, 3.3-fold, and 2.1-fold following the supplementation of RYR, PCO, and RYR + PCO, respectively (
Figure 2
D). However, a significant 2.1-fold (*p* < 0.001) better effect on LDL-C reduction was noticed in the PCO group compared to the RYR group.

Consistently, the dietary supplementation of PCO and RYR + PCO exhibited a substantial (*p* < 0.001) effect in elevating the HDL-C/TC (%) and reducing TG/HDL-C levels compared to the HC- and RYR-supplemented groups (Figure 2E,F).

### 2.3. Plasma Antioxidant Status and Glucose Level

The HC groups displayed substantially compromised plasma FRA and PON activity, indicating the adverse effect of the HC diet on the plasma FRA and PON activity (Figure 3A,B). Supplementation of RYR had a non-significant effect on the HC-alleviated plasma FRA and PON activity. In contrast to this, PCO consumption effectively elevated FRA and PON activity by 1.3-fold and 2.1-fold, respectively, as compared to the HC-supplemented group (Figure 3A,B). Likewise, RYR in the presence of PCO also displayed a substantial effect in improving the FRA and PON activity, which was observed to be ~1.2-fold (*p* < 0.05) better than the respective activities observed in the RYR + HC group.

Similar to the FRA and PON activity, the lowest plasma sulfhydryl content was observed in the HC group, which was not improved following the supplementation of RYR (Figure 3C). The consumption of PCO substantially (*p* < 0.01) elevated the HC-diminished sulfhydryl content by 37%. Also, PCO, in combination with RYR, displayed a 30% (*p* < 0.05) enhanced sulfhydryl content than that observed in the HC and RYR + HC groups.

The plasma glucose level was elevated by HC consumption, which was significantly reduced to 23% and 14% following the supplementation of PCO and RYR + PCO (Figure 3D). However, the supplementation of only RYR fails to diminish any significant reduction in HC-heightened blood glucose levels.

### 2.4. Morphology and Organ/Body Weight Analysis

The morphological analysis of the liver showed a slightly bigger liver size in the HC group. Likewise, the supplementation of RYR showed an enlargement in liver size (Figure 4A). In contrast, the supplementation of PCO and RYR + PCO displayed substantial protection of the liver morphology, which was altered by the consumption of the HC diet. Consistent with the morphological alteration, liver/body weight also diminished among the PCO and RYR + PCO-supplemented groups; however, the difference is non-significant (*p* > 0.05) compared to the HC group (Figure 4F). The kidney morphology slightly changed in the HC group, which was further worsened by the supplementation of RYR (Figure 4B). The supplementation of PCO showed a substantial protective effect against HC-induced nephromegaly; notably, in the presence of PCO, the aggravative adverse effect of RYR on the HC-induced kidney morphology was effectively prevented. Consistent with the morphological outcomes, the kidney/body weight (%) showed a substantial reduction of 40.3% and 36.7% in the PCO- and RYR + PCO-supplemented groups, respectively, compared to the HC group (Figure 4G).

The brain morphology analysis showed a substantial shrinkage of the brain size in the HC- and RYR-supplemented groups. A noticeable alteration appeared in the cerebellum (indicated by red arrow) and in the medulla region (indicated by blue arrow) of the brain obtained from the HC- and RYR-supplemented groups (Figure 4C). The dietary supplementation of PCO and RYR + PCO prevents the HC-induced morphological alteration in the brain. Despite the substantial morphological alteration, non-significant (*p* > 0.05) changes in the brain/body weight (%) were noticed between the different groups (Figure 4H).

The testis and ovary morphology and weight changes showed a non-substantial difference between the groups; however, slightly shrunken testis morphology was noticed in the HC and RYR groups (Figure 4D,E,I,J). 

### 2.5. Liver Histology and Inflammation

H&E staining revealed a substantial neutrophil infiltration in the HC group (Figure 5A,B,F). The supplementation of RYR did not have any preventive effect on the HC-induced neutrophil infiltration; even the neutrophil counts in the RYR group were slightly higher than that observed in the HC group. The supplementation of PCO displayed reduced neutrophil counts compared to the neutrophil counts observed in the HC group. Moreover, PCO was found effective in preventing RYR + HC-induced liver damage, as reflected by a significantly (1.6-fold (*p* < 0.01)) reduced neutrophil count in the RYR + PCO group compared to the RYR-only group.

Consistent with the hepatic H&E results, ORO staining revealed a substantial lipid accumulation in the hepatic tissue of the HC group (Figure 5C,G). The HC-induced lipid accumulation was significantly enhanced by 2.8-fold in the presence of RYR, testifying to the additive effect of RYR on the HC-provoked lipid accumulation. In the PCO-supplemented group, a 6.5-fold lower ORO-stained area was quantified against that observed in the HC group. Also, PCO supplementation prevents RYR-augmented lipid accumulation, as evidenced by a significant 15% diminished ORO-stained area in the RYR + PCO group. The results strengthen the hepatoprotective nature of PCO against the toxicity posed by RYR and HC.

Consistent with the hepatic histological outcomes, IHC staining revealed a substantial IL-6 production in the HC group that was 2.5-fold (*p* < 0.001) higher than in the RYR-supplemented group, attesting to the provocative effect of RYR on the IL-6 level (Figure 5D,E,H). The supplementation of PCO effectively mitigated IL-6 production reflected by 2.4-fold and 1.6-fold reduced IL-6 level as compared to the HC and RYR + HC supplemented groups, respectively.

### 2.6. ROS Production, Apoptosis, and Senescence-Associated β-Galactosidase (SA-β-Gal) Staining

Dihydroethidium (DHE) and acridine orange (AO) staining revealed substantially higher ROS production and apoptosis, respectively, in the HC group (Figure 6A,B,D,E). The supplementation of RYR showed a non-protective effect on the HC-induced ROS generation and apoptosis, as depicted by the non-significant changes in the DHE and AO fluorescent intensity. Contrary to this, the supplementation of PCO effectively counters HC-induced ROS and apoptosis, as evidenced by a significantly (2.2-fold and 2.6-fold) diminished DHE and AO fluorescent intensity in the PCO-supplemented group than the HC group. Also, a significant ~1.8-fold (*p* < 0.001) reduced DHE and AO fluorescence in the RYR + PCO-supplemented group was noticed compared to the RYR-only group, documenting the impact of PCO in mitigating RYR + HC-induced challenges.

A high prevalence of senescence-positive cells was observed in the HC group, which accounts for 18.2% of the senescence-stained area (Figure 6C,F). Supplementation of RYR showed a non-substantial effect against the HC-induced cellular senescence. Contrary to this, PCO consumption effectively inhibits cellular senescence, evidenced by the ~2-fold reduced SA-β-gal-stained area compared to the HC group. Also, RYR, in combination with PCO, significantly inhibits cellular senescence, which was ~1.4-fold lower than the senescent area that appeared in the HC and HC + RYR groups, respectively. 

### 2.7. Hepatic Damage Biomarkers in Plasma

An elevated level of plasma hepatic function biomarkers AST (915.9 IU/L) and ALT (842.5 IU/L) was observed in the HC group, which substantially reduced to 723.4 IU/L (*p* < 0.001) and 629.4 IU/L (*p* < 0.001) by the supplementation of PCO (Figure 7A,B). The supplementation of RYR displayed a non-significant effect in reducing the HC-elevated AST and ALT levels. However, RYR in the presence of PCO reduced the AST level by 13.2% and 8.1% and the ATL level by 16.8% and 10.2% compared to the HC and RYR + HC groups, respectively.

### 2.8. Histological Analysis of Kidneys

The kidney histology outcome suggests a disorganized proximal and distal tubular structure with the frequent occurrence of elevated tubular lumen (indicated by a red arrow), and luminal debris in the tubular cast (indicated by a blue arrow) of the kidney section of the HC group (Figure 8A). The supplementation of RYR did not have a kidney-protective effect, as shown by the highly disorganized proximal and distal tubule with elevated tubular lumen and luminal debris. PCO supplementation showed the restoration of HC-induced kidney impairment, though elevated tubular lumen and luminal debris were noticed occasionally. RYR, in combination with PCO, displayed a protective effect on HC-induced kidney impairment; however, the presence of luminal debris was noticed in several places.

The DHE and AO fluorescent staining depicts the high prevalence of ROS and apoptosis in the HC- and RYR-supplemented groups. The consumption of PCO alone or in combination with RYR was found to be effective in inhibiting ROS production and apoptosis, as reflected by the 3.1-fold and 1.7-fold reduced DHE and 3.5-fold and 1.8-fold reduced AO fluorescence intensity compared to the HC group, respectively (Figure 8B,C,E,F).

Likewise, a high prevalence of senescent cells was noticed in the HC group, which was significantly augmented by 8.3% (*p* < 0.05) by the supplementation of RYR. Unlike this, in the PCO-supplemented group, a significant ~2-fold reduced senescent area was quantified compared to the senescent area in the HC- and RYR + HC-supplemented groups (Figure 8D,G). Also, 1.6-fold reduced cellular senescence was noticed in the RYR + PCO-supplemented group compared to the RYR-only group.

### 2.9. Histological Analysis of the Brain

The H&E staining revealed substantial vacuolation and mononuclear cells with a clear zone in the optic tectum (TeO) and periventricular gray zone (PGZ) in the brain of the HC group that became worse along with the occasional presence of hyperemia following RYR supplementation (Figure 9A). On the contrary, the consumption of PCO displayed substantial protection by effectively reducing the vacuolation and mononuclear cells with clear zones in the TeO and PGZ regions. Similarly, RYR in the presence of PCO showed a protective role against HC-induced brain impairment; however, hyperemia was noticed in certain areas. 

The DHE and AO staining suggested a high amount of ROS and apoptosis around the PGZ, mainly condensed in the valvular cerebella (Val), in the HC group (Figure 9B,C). Likewise, analogous DHE and AO fluorescent intensity was observed in the RYR group, suggesting that RYR did not affect HC-induced ROS and apoptosis. The co-supplementation of PCO effectively reduced the HC-elevated DHE fluorescent intensity by 2-fold (Figure 9B,G) and AO fluorescent intensity by 1.8-fold (Figure 9C,H) compared to the HC group, attesting to the substantial potential of PCO to inhibit ROS production and apoptosis. Also, RYR in the presence of PCO showed a ~1.7-fold reduction in the DHE and AO fluorescent intensity compared to the HC and HC + RYR groups.

A substantial amount of senescence-positive cells around the vascular lacuna area of the postrema (Vas) beneath the tectal ventricle (TeV) and across the PGZ region was noticed in the brain section of the HC- and RYR-supplemented group (Figure 9D–F). The supplementation of PCO alone and in combination with RYR efficiently reduced the HC-induced cellular senescence, reflected by the 1.8-fold and 1.4-fold reduced SA-β-gal-stained area compared to the HC group (Figure 9I).

### 2.10. Histological Analysis of the Testis

The H&E staining of the testis revealed loosely organized seminiferous tubules, void space in the lumen, and minor ruptures in the lamina basal membrane with a 15.6% interstitial space between the seminiferous tubules in the HC group (Figure 10A,B,F). These changes are persistent in the RYR-supplemented group, with frequent ruptures in the lamina basal membrane and heightened interstitial space between the seminiferous tubules, which was significantly (34.4%) higher than that noticed in the HC group. The supplementation of PCO prevents testis impairment, marked by a 2-fold (*p* < 0.001) reduced interstitial space between seminiferous tubules compared to the HC group. RYR, in combination with PCO, showed slightly better testis protection than the RYR-only group; however, a rupture in the lamina basal membrane and void space in the lumen were intermittently observed.

DHE (Figure 10C,G) and AO (Figure 10D,H) fluorescent staining revealed higher ROS generation and apoptosis in the RYR-supplemented group, which is similar to the DHE and AO fluorescent intensity observed in the HC group. Unlike this, the supplementation of PCO substantially dismisses the HC-induced ROS level and apoptosis, as reflected by the 2-fold (*p* < 0.001) and 3-fold (*p* < 0.001) reduced DHE and AO fluorescent intensity compared to the HC group. Similarly, in the presence of RYR + PCO, a substantially 1.9-fold (*p* < 0.001) and 1.6-fold (*p* < 0.001) reduced DHE and AO fluorescent intensity was noticed compared to the RYR-supplemented group, testifying the effect of PCO in preventing RYR + HC-induced ROS production and apoptosis.

The SA-β-gal staining suggests higher cellular senescence in the HC- and RYR-supplemented group that was significantly reduced by 2-fold (*p* < 0.001) and 1.5-fold (*p* < 0.001) in the PCO- and RYR + PCO-supplemented groups as compared to the HC and RYR + HC groups, respectively (Figure 10E,I).

### 2.11. Histological Analysis of the Ovary

The ovary H&E staining revealed minor histological changes in the ovarian tissue with a high prevalence of pre-vitellogenic oocytes in the HC group (Figure 11A,E). The supplementation of PCO displayed significantly (7% (*p* < 0.01)) reduced pre-vitellogenic oocyte counts than that observed in the HC group, suggesting the beneficial impact of PCO against the HC-imposed ovary damage. Contrary to PCO, no beneficial effect of RYR and RYR + PCO was noticed on the HC-alleviated pre-vitellogenic oocytes. Interestingly, among all the groups, a non-significant difference between the early and mature vitellogenic oocyte counts was noticed.

DHE (Figure 11B,F), AO (Figure 11C,G), and SA-β-gal (Figure 11D,H) staining revealed a higher extent of ROS, apoptosis, and senescence in the RYR-supplemented group that was analogous with changes observed in the HC group. On the contrary, the supplementation of PCO displayed substantially reduced DHE, AO, and SA-β-gal-stained area that was ~2-fold (*p* < 0.001) lower than the HC group. Consistently, the RYR + PCO supplementation inhibited ROS generation, apoptosis, and senescence, marked by a ~1.5-fold reduced DHE, AO, and SA-β-gal stained area compared to the RYR-supplemented group.

## 3. Discussion

The current study evaluated the comparative effects of PCO and RYR (*Beni-koji* Coleste help) and their combination (RYR + PCO) on HC-induced dyslipidemia and altered antioxidant variables of zebrafish. *Beni-koji* RYR (Kobayashi Co., Ltd., Osaka, Japan) was selected for its established cholesterol-lowering properties [25] and the controversies associated with its safety and efficacy [31]. While most of these studies are based on human subjects consuming RYR, there is a notable lack of animal studies assessing the long-term effect of RYR supplementation, particularly considering recent controversies about hidden toxicity. This study addresses this gap by supplementing RYR for 12 weeks using the zebrafish as a model organism. The selection of zebrafish as a model organism in the present research was based on its highly similar lipid metabolism profile to humans [32], as it harbors most of the critical enzymes and receptors involved in human lipid metabolism, supporting the applicability of zebrafish data to the human health context.

High cholesterol consumption is associated with dyslipidemia and several other diseases [33,34,35]. Also, the survivability of zebrafish is severely compromised by the consumption of high cholesterol. Consistently, in the present study, compromised zebrafish survivability was noticed in response to HC consumption, which was enhanced by the supplementation of RYR, suggesting the additive/adverse effect of RYR on zebrafish mortality. These results are aligned with reports showing the fatal effect of RYR consumption [31,36]. Particularly, in early 2024, the Kobayashi RYR scandal was reported, with five deaths and more than two-hundred people hospitalized following the Kobayashi *Beni-koji* RYR supplement, which posed a serious safety concern about RYR-based supplements [31]. Later, it was noticed that the toxicity is primarily associated with the contamination of puberulic acid, produced by blue mold [31]. Due to these adverse events, Kobayashi Pharmaceuticals stopped the production of *Beni-koji* RYR and withdrew all RYR products from the global market [37]. Despite this, some concern about RYR’s adverse events and toxic reactions have been noticed in clinical practice [28,29].

In the present study, contrary to RYR, the consumption of PCO effectively prevented HC-induced zebrafish mortality. Moreover, the consumption of RYR together with PCO displayed improved zebrafish survivability, underscoring the substantial protective effect of PCO against RYR + HC, which triggered severe mortality. These study outcomes are supported by earlier studies documenting the protective effect of PCO against a variety of stressors, including carboxymethyllysine (CML) and high-cholesterol-induced toxicity [38,39].

Consumption of both RYR and PCO was found effective in countering HC-induced dyslipidemia. However, compared to RYR, a much higher efficacy of PCO was observed, specifically in elevating HDL-C and minimizing the LDL-C level. The study outcomes corroborate earlier findings depicting the pharmacological effects of RYR [25] and PCO [10] against dyslipidemia. The curative mechanism of = RYR against dyslipidemia is mainly due to the presence of monacolin K in it, which is well known to inhibit 3-hydroxy-3-methyl-glutaryl-coenzyme A (HMG-CoA) reductase, a main rate-limiting enzyme of the cholesterol biosynthesis pathway [26]. Similarly, PCO also has an inhibitory impact on HMG-CoA reductase [11,40]. Also, hexacosanol (a major LCAA of PCO) transcriptionally regulates the expression of cholesterol biosynthetic genes by inhibiting the nuclear translocation of sterol regulatory element-binding protein (SREBP)-2 [41]. In addition, the established effect of PCO on improving the cholesterol efflux capacity and conversion of hepatic cholesterol to bile acid and its subsequent removal from feces has been reported as an important event in countering dyslipidemia [18]. In addition, a noteworthy positive effect of PCO has been observed to inhibit the cholesteryl ester transfer protein (CETP) [15] that eventually elevates the HDL-C level. PCO consumption has also been associated with reducing apolipoprotein (Apo)-B expression and elevation of the LDL-C receptor in the liver, which has a positive modulatory effect on the blood LDL-C level [12]. In a study with lovastatin, simvastatin, pravastatin, and atorvastatin, PCO demonstrated comparable lipid-lowering efficacy [19], attesting its comparative efficacy with the classical statins. Additionally, in individuals with type 2 diabetes and hypercholesteremia, PCO was shown to outperform lovastatin in reducing the LDL/HDL ratio and augmenting HDL levels, with significantly fewer side effects than lovastatin [19].

The HC diet provoked diminished plasma FRA and PON activity, and sulfhydryl content remained unchanged in response to RYR consumption. On the contrary, PCO consumption effectively reversed the HC-induced changes in the plasma antioxidant. The outcomes are in accordance with previous studies documenting the impact of PCO as a cellular antioxidant that effectively elevated the plasma FRA and PON activity [17]. Plasma sulfhydryl content is an important stress marker [42,43], and its diminished level has been recognized in various pathological conditions [44]. PCO showed a substantial effect in elevating the HC-diminished sulfhydryl content, while RYR consumption failed to do so. The combined results of the FRA and PON activity and the plasma sulfhydryl contents establish the remarkable protective effect of PCO against HC-induced adverse events.

It has been documented that high cholesterol and fat consumption is associated with elevated blood glucose levels [45] and is considered one of the key factors for the onset of type 2 diabetes [46]. Furthermore, an association between high cholesterol and insulin resistance has been documented [47], which eventually impacts blood glucose levels. Herein, an elevated blood glucose level was noticed in the HC group, which was substantially prevented by the consumption of PCO. Similarly, the HC-induced elevation in blood glucose levels was notably diminished in zebrafish supplemented with RYR + PCO. However, the glucose levels in the RYR + PCO group remained higher than those supplemented with PCO alone. This difference is likely attributed to the lower PCO concentration (0.5%) compared to the higher amount (1.0%) in the PCO group, indicating a potential dose-dependent effect of PCO in mitigating HC-induced hyperglycemia. Unlike this, RYR displayed no effect in reducing the HC-induced blood glucose level. The hypoglycemic effect of PCO is compliant with earlier studies describing the positive impact of PCO on the blood glucose level in zebrafish and human subjects [48]. The affirmative role of PCO in insulin sensitization and secretion [48] is among the key aspects of its hypoglycemic effect. Also, a substantial effect of PCO on the activation of phosphatidylinositol 3-kinase/protein kinase B (PI3K/Akt) signaling has been recognized [48], which serves as a key regulator of insulin sensitivity. Furthermore, studies deciphered the inhibitory effect of octacosanol (a major LCAA of PCO) towards α-glucosidase [49], a key enzyme responsible for the conversion of complex carbohydrates into simple sugar [50], which strengthens the current outcomes of PCO’s hypoglycemic effect.

HC consumption is well recognized to cause fatty liver changes and hepatic inflammation [51]. Consistently, we have also noticed a substantial effect of HC consumption on hepatic histology and fatty liver changes. Intriguingly, the co-consumption of RYR augments HC-triggered hepatic damage, marked by significantly higher neutrophil infiltration, lipid deposition, and production of proinflammatory cytokine IL-6. These findings are in agreement with a report indicating the potential of RYR to cause fatty liver changes (steatosis) and inflammation [36]. In contrast to RYR, PCO effectively mitigated HC-induced liver damage, which is consistent with earlier findings describing PCO’s hepatoprotective role against a variety of external stresses [38,52]. Notably, the presence of hexacosanol (an LCAA) in PCO has been known to induce autophagy through the upregulation of an autophagy-related gene (ATG16L) and anti-microtubule-associated 1A/1B light chain (LC3II) [41] that induce healing events to prevent fatty liver changes [41,53].

In addition to fatty liver changes, HC consumption has been documented to provoke excessive ROS generation and apoptosis [54]. Similar to the hepatic histology finding, PCO effectively subdued HC-induced ROS and apoptosis, while RYR failed to counter this. The cellular antioxidant nature of PCO is the main reason for the diminished hepatic ROS level, which eventually prevents apoptosis, as ROS-induced oxidative stress is a key culprit behind apoptotic cell death [55]. Also, PCO consumption has been observed to inhibit caspase 3 expression in the rat liver [52] and thus has an anti-apoptotic effect. Likewise, the low cellular senescence in response to PCO is correlated with lower ROS production, as the induction of cellular senescence is directly correlated with ROS-triggered oxidative stress [56,57].

The accumulating literature suggests an adverse impact of high cholesterol on the kidneys [58] and reproductive organs [59,60]. Similarly, herein, we have noticed an adverse effect of 12 weeks of HC intake on zebrafish’s kidney and reproductive organs, which were effectively neutralized by the co-consumption of PCO. Unlike PCO, RYR displayed no curative effect; rather, it aggravated the HC-induced kidney impairment. These findings are supported by reports documenting the intense kidney-damaging effect of RYR [31]. A recent report in 2024 revealed that the Kobayashi RYR scandal was associated with acute renal injury following *Beni-koji* consumption, testifying its adverse effect on the kidney and supporting the outcomes of the present study [31]. Interestingly, supplementation with PCO effectively mitigated the adverse effects of RYR, emphasizing its substantial kidney-protective role against RYR- and HC-induced impairment. Moreover, elevated ROS levels, apoptosis, and cellular senescence were observed in the kidneys of zebrafish exposed to HC and RYR, whereas these markers were markedly reduced in the group supplemented with PCO. We speculated that the cellular antioxidant effect of PCO [38] is the key contributor towards diminished ROS production that also impacts apoptosis and senescence. This notion is supported by earlier reports describing the association of ROS-induced oxidative stress with the induction of apoptosis [55] and senescence [56,57].

PCO showed substantial protective effects on the liver and kidney, subsequently impacting brain health. This notion is supported by reports depicting a tight association of the liver–brain axis [61] and kidney–brain axis [62,63]. Excessive fat deposition in the liver modifies the composition and levels of fat-derived products in circulation, disrupting the blood–brain barrier and facilitating the accumulation of toxic substances and inflammatory cells in the brain, ultimately causing brain damage [64]. Similarly, oxidative stress in chronic kidney disease contributes to brain lesions and cognitive decline, underscoring the interplay between kidney and brain health [62] and potentially leading to neuropsychiatric disorders.

Also, PCO exhibits a protective effect on the testes and ovaries against the harmful impact of HC. In contrast, RYR intake induces acute toxicity in the reproductive organs, which is substantially mitigated by the co-supplementation of PCO. Research on the influence of PCO on reproductive organ functionality remains limited; however, one such study reported that PCO consumption positively influenced zebrafish egg-laying behavior and improved the survival rate of fertilized eggs [39]. Similarly, studies on the effects of RYR on reproductive organs are spare, with one report highlighting the adverse effect of RYR on erectile dysfunction and reduced libido [65]. We believe that the better testis- and ovary-protective role of PCO was due to its cellular antioxidant nature that substantially impacts ROS production. This perspective is strongly supported by earlier reports deciphering the importance of antioxidants in protecting the testis [66] and ovary [67] against various stimulations.

## 4. Materials and Methods

### 4.1. Materials

Policosanol 20 tablets (198 mg), containing Cuban sugarcane wax alcohols, were provided by RAYDEL Australia Pty, Ltd., (Thornleigh, NSW, Australia). Red yeast rice extract tablets (200 mg) were purchased from Kobayashi Pharmaceutical Co., Ltd., Osaka, Japan. All the other chemicals and reagents else stated are of analytical grade and used as supplied. A detailed list of the reagents and chemicals used is provided in Appendix A.

### 4.2. Zebrafish Aquaculture

Zebrafish were maintained in an aerated water tank with a circulating water supply following the standard guidelines of the Animal Care and Committee and approved by Raydel Research Institute (RRI, code of approval RRI-23-007, 27 July 2023). A water temperature of 28 °C ± 1.0 °C and a periodic light (14 h) and dark (10 h) cycle were maintained through the culturing period.

### 4.3. Preparation for Different Diets

A normal tetrabit diet (ND), a regular zebrafish diet, was used as a base to form the different diet formulations. The ND was mixed with cholesterol to prepare a high-cholesterol diet (HC). To prepare for the HC diet, 250 g of ND was infused with 10 g of cholesterol (equivalent to 4%, wt/wt) and mixed properly using a glass rod. Subsequently, chloroform was added to the ND-infused cholesterol and physically agitated for proper cholesterol distribution. Finally, the chloroform phase was evaporated in a fume hood to obtain ND-infused chloroform, named the HC diet. To ascertain the appropriate distribution of cholesterol, a 25 g sample (HC) was mixed with chloroform to extract the bound cholesterol. The chloroform phase was separated following evaporation to obtain the extracted cholesterol, which was quantified using a commercial cholesterol diagnostic kit (cholesterol, T-CHO, Clean TS-S; Walko Pure Chemicals, Osaka, Japan).

The HC was mixed with ground red yeast rice extract tablets (RYR, final 1.0%, wt/wt) or ground policosanol tablets (PCO, final 1.0%, wt/wt) or with RYR (final 0.5%, wt/wt) + PCO (final 0.5%, wt/wt) to make three different dietary formulations, abbreviated as HC + 1.0% RYR, HC + 1.0% PCO, and HC + 0.5% (RYR + PCO), respectively.

The 1.0% amount was selected based on the preliminary experiment, where we supplemented PCO (0.5–5%) to hyperlipidemic zebrafish and further assessed the effect on the plasma lipid profile and hepatic biomarkers (AST and ALT). The results outlined the increasing protective effect of PCO up to 1.0%; afterwards (>1.0% PCO), the effect was nearly similar. Therefore, 1.0% of PCO was selected as the smallest amount that imparted the most desirable effect. An equivalent amount of RYR (1.0%) was applied to ensure similar conditions for the comparative evaluation.

### 4.4. Supplementation of Different Diets to Zebrafish

Adult zebrafish (~16 weeks old) were fed with an HC diet for eight weeks to develop hyperlipidemic zebrafish. The hyperlipidemic zebrafish (*n* = 112) were segregated into four cohorts (*n* = 28/group). In group I, zebrafish were exclusively fed with HC, while the zebrafish in groups II, III, and IV were fed with HC + 1.0% RYR, HC + 1.0% PCO, and HC + 0.5% (RYR + PCO), respectively. For 12 weeks, the respective diets (10 mg/zebrafish) were fed to each group two times a day (morning and evening).

The survivability of zebrafish among all the groups was measured every day until 12 weeks of supplementation. The body weight of zebrafish in each group was measured gravimetrically on the starting day (day 0) and on the final day (12 weeks). For body weight measurement [at day 0 and final day (12 weeks)], zebrafish from the respective groups were anesthetized by submerging them into a 2-phenoxyethanol solution (0.1%). The anesthetized zebrafish were removed from the solution, kept on a tissue paper surface, and immediately weighed using a electronic weigh balance (Ohaus, Parsippany-Troy Hills, NJ, USA).

### 4.5. Euthanizing Zebrafish and Collection of Blood and Organs

At the end of 12 weeks of supplementation, the zebrafish from the different groups were euthanized using hypothermic shock and immediately processed to collect blood samples. Zebrafish were dissected under an optical microscope (10× magnification) to obtain different organs (liver, kidney, brain, testis, and ovary), which were individually weighed by using a Mettler Toledo MA95 Analytical Balance (Mettler-Toledo GmbH, Greifensee, Switzerland), and then preserved in 10% formalin.

### 4.6. Estimation of Blood Antioxidant Variables

The plasma antioxidant status among the different groups was assessed by quantifying the ferric ion reduction ability (FRA), paraoxonase activity, and plasma sulfhydryl content as per previously described methods [68]. In brief, plasma (20 μL, 0.1 mg/mL equivalent protein) was mixed with 180 μL of FRA reagent [68]. Following 60 min incubation at room temperature, the absorbance at 593 nm was recorded. For the estimation of PON activity, plasma (40 μL, 0.1 mg/mL equivalent protein) was blended with 160 μL of paraoxon-ethyl (0.55 M). After 60 min incubation at 25 °C, the absorbance at 415 nm was recorded, and the PON activity (μU/L/min) was quantified using the 17,000 M^−1^ cm^−1^ molar extinction coefficient (ε) of *p*-nitrophenol (product).

The sulfhydryl content in plasma (50 μL, 0.1 mg/mL equivalent protein) was determined by mixing the sample with an equal volume of 5,5-dithiol-bis-(2-nitrobenzoic acid) (DTNB), followed by 12 h incubation at room temperature [68]. Finally, the absorbance 412 nm was determined, and the plasma sulfhydryl (mmol/mg protein) was quantified using the 13,600 M^−1^ cm^−1^ molar absorbance coefficient (ε) of DTNB. 

### 4.7. Estimation of Blood Glucose Level, Lipid Profile, and Hepatic Function Biomarkers

The blood glucose level was assessed by an automated blood glucose meter (AccuCheck, Roche, Basel, Switzerland). The total blood cholesterol (TC), triglyceride (TG), high-density lipoprotein cholesterol (HDL-C), aspartate aminotransferase (AST), and alanine aminotransferase (ALT) were quantified using commercial kits following the manufacturer’s instructions. A detailed methodology is provided in Appendix A. 

### 4.8. Histological Analysis and Immunohistochemistry (IHC)

Sections (7 μm thick) of different organs (liver, kidney, brain, testis, and ovary) were obtained using a cryo-microtome (Leica Cm 1510S, Leica Biosystem, Nussloch, Germany). For the histological analysis, different organs were stained with hematoxylin and eosin (H&E) [69] and subsequently visualized under a binocular microscope (Motic microscopy PA53MET, Hong Kong, China) equipped with a digital camera.

For the oil red O staining (ORO) [70], the hepatic section (7 μm thick) was covered with ORO solution (250 μL, 3 mg/mL) for 10 min and subsequently washed with water and visualized under a microscope.

The interleukin (IL)-6 level in the hepatic section (7 μm thick) was determined by IHC [71]. The tissue section was incubated for 24 h with IL-6-specific primary antibody (ab9324, Abcam, London, UK) followed by section development using an EnVision + System-HRP polymer kit (Code K4001, Dako, Glostrup, Denmark) containing secondary antibody against the primary immunoglobulin IL-6.

### 4.9. Imaging for Reactive Oxygen Species (ROS), Apoptosis, and Senescence

The ROS generation and cellular senescence were determined by dihydroethidium (DHE) [72] and acridine orange (AO) [73] fluorescent staining. Briefly, the tissue section (7 μm thick) was covered with the DHE (30 μM) and AO (5 mg/mL) solutions. After 30 min, the tissue section was rinsed and visualized under a fluorescent microscope at excitation/emission wavelengths of 585 nm/615 nm (for DHE) and 505 nm/535 nm (for AO).

The cellular senescence was detected by senescence-associated β galactosidase staining [74]. In brief, the tissue section (7 μm thick) was covered with 5-bromo-4-chloro-3-indolyl-β-D-galactopyranoside (X-gal, 1 mg/mL). After 16 h incubation, the stained section was washed and subsequently visualized under the microscope to detect the blue-stained senescence-positive cells.

### 4.10. Statistical Analysis

The data obtained from three independent experiments are represented as the mean ± SEM. The statistical divergence between the groups was established by one-way analysis of variance (ANOVA) followed by Tukey’s post hoc analysis and Student’s *t*-test (for pairwise comparison) using the SPSS software (version 29.0; Chicago, IL, USA).

## 5. Conclusions

PCO outplays RYR to counter HC-elevated dyslipidemia in zebrafish with a notable elevation of HDL-C level and HDL-associated PON activity. In addition, PCO improved the plasma antioxidant status, elevated sulfhydryl content, and diminished the HC-augmented blood glucose level. RYR triggered hepatic IL-6 production, prompted fatty liver changes, and caused damage to the kidneys and reproductive organs by inducing ROS, apoptosis, and cellular senescence, resulting in high zebrafish mortality. These adverse events were effectively mitigated by PCO consumption. Also, PCO protects the zebrafish brain from oxidative damage and minimizes apoptosis and senescence. The findings underscore RYR’s severe toxicity and establish PCO’s beneficial potential to counter dyslipidemia and associated detrimental events among the different organs.

## Figures and Tables

**Figure 1 pharmaceuticals-18-00200-f001:**
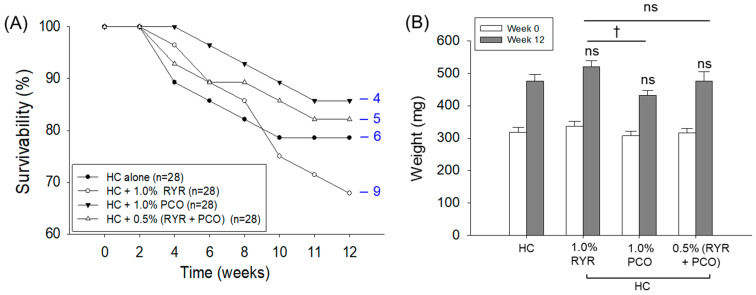
A comparative effect of red yeast rice (RYR) and policosanol (PCO) supplementation on survival and body weight of zebrafish fed a high-cholesterol diet. (**A**) Kinetics of zebrafish survivability over 12 weeks; numerical values (blue fonts) depict dead zebrafish in the respective groups. (**B**) Zebrafish body weight. HC represents the high-cholesterol diet, HC + 1.0% RYR or 1.0% PCO represents the high-cholesterol diet supplemented with 1.0% red yeast rice or 1.0% policosanol, and HC + 0.5% RYR + 0.5% PCO represents the high-cholesterol diet supplemented with 0.5% each of red yeast rice and policosanol. ^†^ (*p* < 0.05) represents the statistical significance compared to the 1.0% RYR group; ns represents a non-significant (*p* > 0.05) difference between the groups for body weight.

**Figure 2 pharmaceuticals-18-00200-f002:**
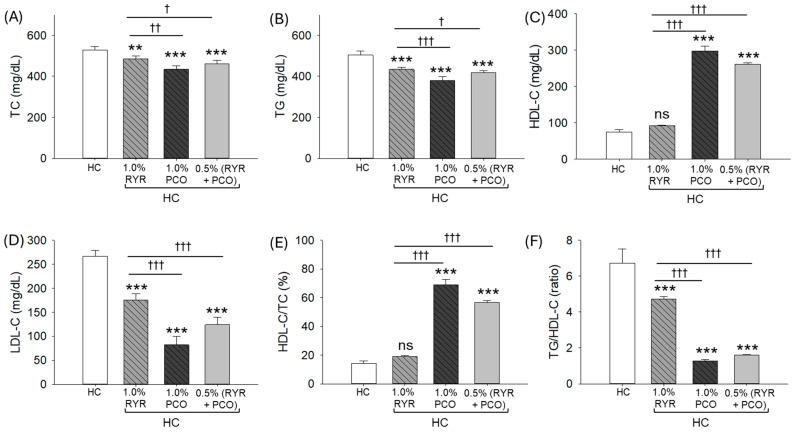
A comparative effect of red yeast rice (RYR) and policosanol (PCO) supplementation for 12 weeks on the plasma lipoprotein profile of the zebrafish fed with a high-cholesterol diet. (**A**) Total cholesterol (TC), (**B**) triglycerides (TGs), (**C**) high-density lipoprotein cholesterol (HDL-C), (**D**) low-density lipoprotein cholesterol (LDL-C), (**E**) percentage HDL-C/TC, and (**F**) ratio of TG/HDL-C. HC represents the high-cholesterol diet, HC + 1.0% RYR or 1.0% PCO represents the high-cholesterol diet supplemented with 1.0% red yeast rice or 1.0% policosanol, and HC + 0.5% RYR + 0.5% PCO represents the high-cholesterol diet supplemented with 0.5% each of red yeast rice and policosanol. Statistical significance is shown at *p* < 0.01 (**) and *p* < 0.001 (***) compared to the HC group, while ^†^ (*p* < 0.05), ^††^ (*p* < 0.01), and ^†††^ (*p* < 0.001) represent the statistical significance compared to the 1.0% RYR group; ns represents a non-significant (*p* > 0.05) difference between the groups.

**Figure 3 pharmaceuticals-18-00200-f003:**
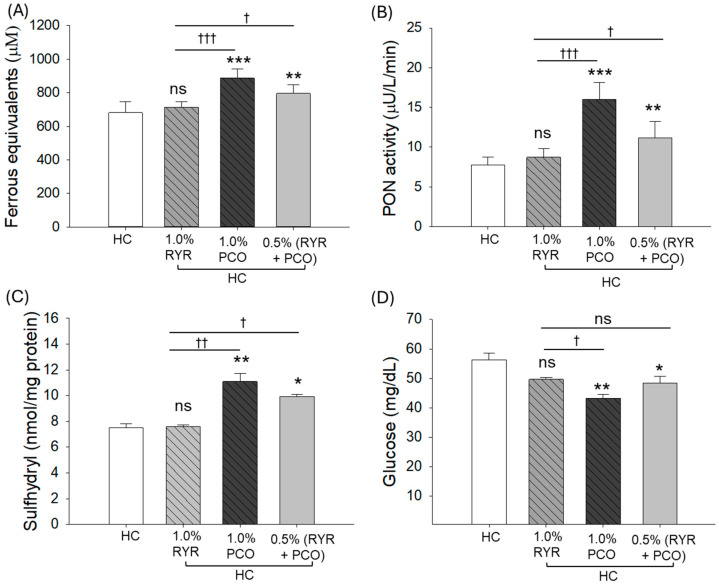
Effect of 12-week supplementation of red yeast rice (RYR) and policosanol (PCO) on the blood antioxidant variables and glucose levels of zebrafish fed a high-cholesterol diet. (**A**) Plasma ferrous ion reduction ability (FRA). (**B**) Paraoxonase (PON) activity. (**C**) Plasma sulfhydryl content. (**D**) Blood glucose level. HC represents the high-cholesterol diet, HC + 1.0% RYR or 1.0% PCO represents the high-cholesterol diet supplemented with 1.0% red yeast rice or 1.0% policosanol, and HC + 0.5% RYR + 0.5% PCO represents the high-cholesterol diet supplemented with 0.5% each of red yeast rice and policosanol. Statistical significance is shown at *p* < 0.05 (*), *p* < 0.01 (**), and *p* < 0.001 (***) compared to the HC group, while ^†^ (*p* < 0.05), ^††^ (*p* < 0.01), and ^†††^ (*p* < 0.001) represent the statistical significance compared to the 1.0% RYR group; ns represents a non-significant (*p* > 0.05) difference between the groups.

**Figure 4 pharmaceuticals-18-00200-f004:**
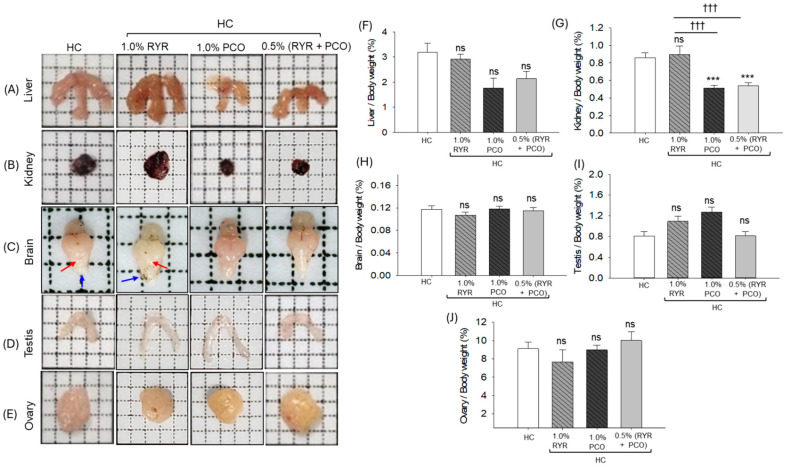
Zebrafish organ morphology following 12-week supplementation of red yeast rice (RYR) and policosanol (PCO) in the presence of high-cholesterol diet. Representative images of (**A**) liver, (**B**) kidney, (**C**) brain, (**D**) testis, and (**E**) ovary. Red and blue arrows in the brain indicate cerebellum and medulla region. (**F**–**H**) depict liver, kidney, and brain weight, respectively, with respect to their body weight. (**I**,**J**) depict testis and ovary weight with respect to body weight. HC represents high-cholesterol diet, HC + 1.0% RYR or 1.0% PCO represents high-cholesterol diet supplemented with 1.0% red yeast rice or 1.0% policosanol, and HC + 0.5% RYR + 0.5% PCO represents high-cholesterol diet supplemented with 0.5% each of red yeast rice and policosanol. Statistical significance at *p* < 0.001 (***) compared to the HC group; ^†††^ (*p* < 0.001) represents statistical significance compared to 1.0% RYR group; ns represents non-significant (*p* > 0.05) difference between groups.

**Figure 5 pharmaceuticals-18-00200-f005:**
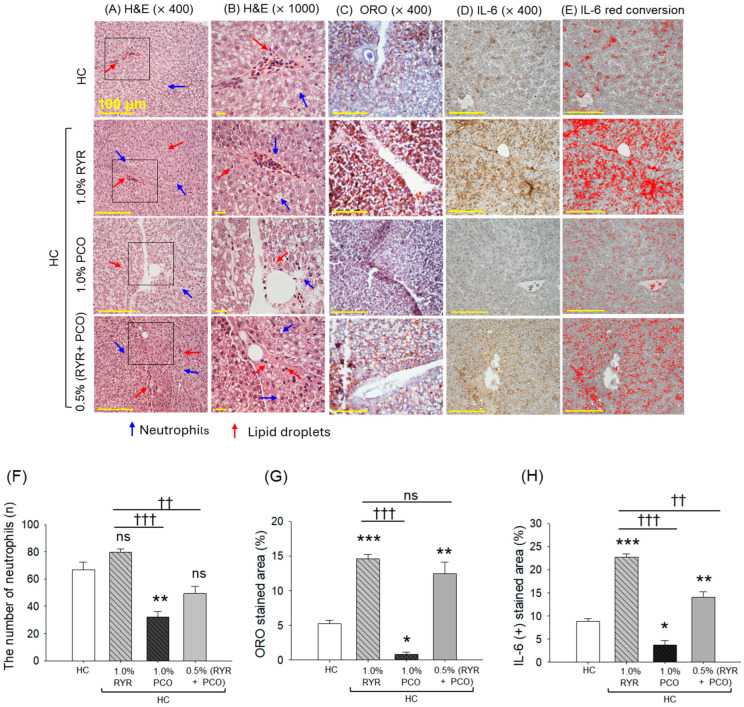
A comparative effect of red yeast rice (RYR) and policosanol (PCO) supplementation for 12 weeks on the hepatic histology of the zebrafish fed with a high-cholesterol diet. (**A**) Hematoxylin and eosin (H&E) staining: blue and red arrows highlight the infiltration of neutrophils and lipid droplets, respectively [100 μm, scale bar]. The H&E area in the black box is 1000× magnified and images (**B**) represents this magnified view. (**C**) Oil red O staining. (**D**) Immunohistochemical (IHC) analysis for the detection of interleukin (IL)-6. (**E**) Image J-based (version 1.53, https://imagej.net/ij, accessed on 16 June 2023) interconversion of brown color to red color (at brown color threshold value 20–120) to enhance the visibility of the IL-6-stained area. (**F**) Percentage neutrophil counts. (**G**) Quantification of ORO-stained area, and (**H**) Quantification of IL-6-stained area. HC represents the high-cholesterol diet, HC + 1.0% RYR or 1.0% PCO represents the high-cholesterol diet supplemented with 1.0% red yeast rice or 1.0% policosanol, and HC + 0.5% RYR + 0.5% PCO represents the high-cholesterol diet supplemented with 0.5% each of red yeast rice and policosanol. † Statistical significance at *p* < 0.01 (*), *p* < 0.01 (**), and *p* < 0.001 (***) compared to the HC group and *p* < 0.01 (^††^), and *p* < 0.001 (^†††^) compared to the 1.0%RYR group. ns represents a non-significant (*p* > 0.05) difference between the groups.

**Figure 6 pharmaceuticals-18-00200-f006:**
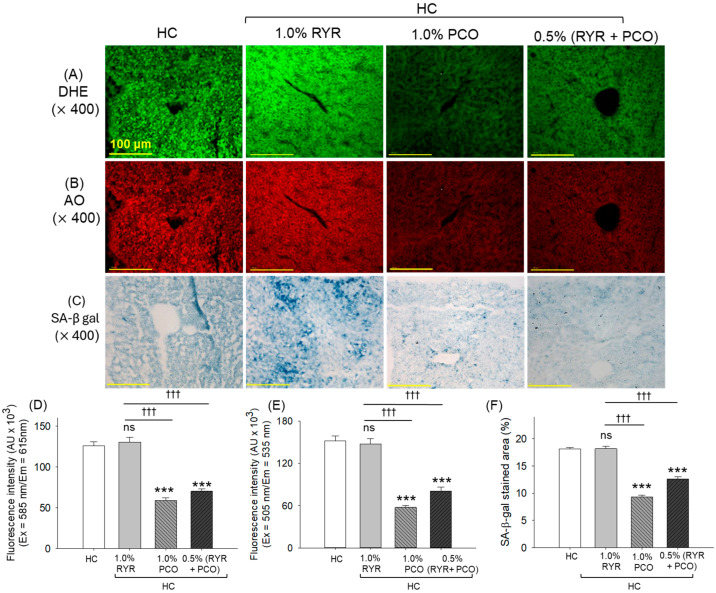
Effect of 12-week supplementation of red yeast rice (RYR) and policosanol (PCO) on the hepatic reactive oxygen species (ROS), apoptosis, and senescence of adult zebrafish fed with high-cholesterol diet feed. (**A**) Dihydroethidium (DHE) and (**B**) acridine orange (AO) fluorescent staining. (**C**) Senescence-associated-β galactosidase (SA-β-gal) staining [100 μm, scale bar]. Quantification of (**D**) DHE, (**E**) AO fluorescent intensities, and (**F**) SA-β-gal-stained area. HC represents the high-cholesterol diet, HC + 1.0% RYR or 1.0% PCO represents the high-cholesterol diet supplemented with 1.0% red yeast rice or 1.0% policosanol, and HC + 0.5% RYR + 0.5% PCO represents the high-cholesterol diet supplemented with 0.5% each of red yeast rice and policosanol. Statistical significance at *p* < 0.001 (***) compared to the HC group; ^†††^ (*p* < 0.001) represents the statistical significance compared to the 1.0% RYR group; ns represents a non-significant (*p* > 0.05) difference between the groups.

**Figure 7 pharmaceuticals-18-00200-f007:**
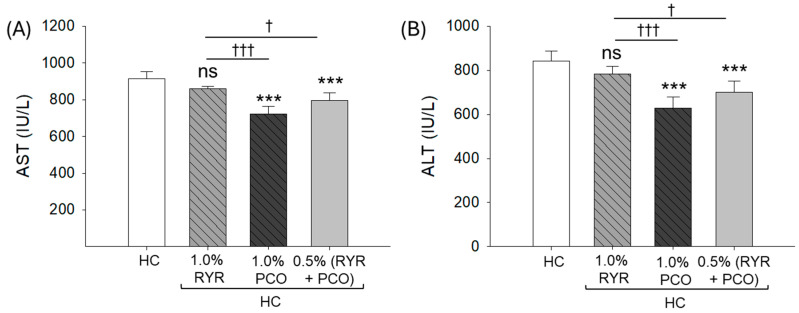
Estimation of the hepatic function biomarkers (**A**) aspartate aminotransferase (AST) and (**B**) alanine aminotransferase (ALT) in the blood of zebrafish following 12 weeks of supplementation of red yeast rice (RYR) and policosanol (PCO) along with a high-cholesterol diet. HC represents the high-cholesterol diet, HC + 1.0% RYR or 1.0% PCO represents the high-cholesterol diet supplemented with 1.0% red yeast rice or 1.0% policosanol, and HC + 0.5% RYR + 0.5% PCO represents the high-cholesterol diet supplemented with 0.5% each of red yeast rice and policosanol. Statistical significance at *p* < 0.001 (***) compared to the HC group; while ^†^ (*p* < 0.05) and ^†††^ (*p* < 0.001) represent the statistical significance compared to the 1.0% RYR group; ns represents a non-significant (*p* > 0.05) difference.

**Figure 8 pharmaceuticals-18-00200-f008:**
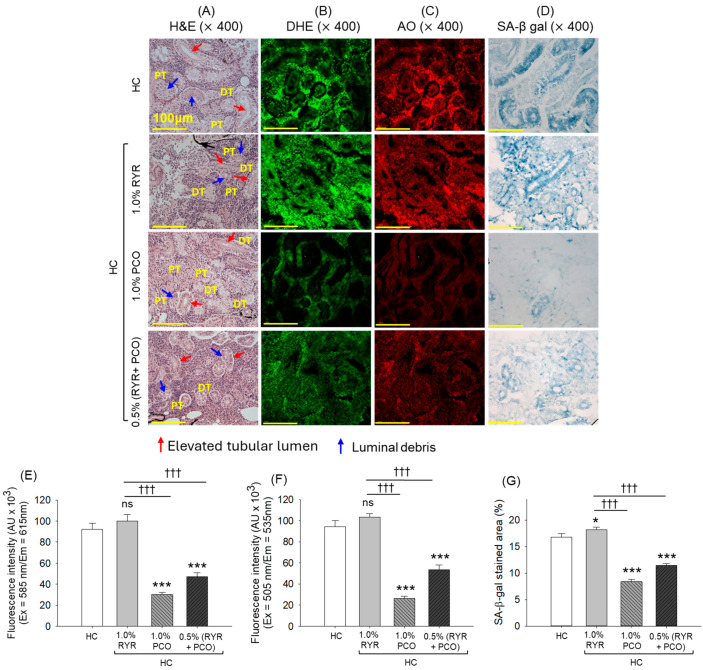
Kidney histology of adult zebrafish following 12 weeks of supplementation with red yeast rice (RYR) and policosanol (PCO) along with a high-cholesterol diet. (**A**) Hematoxylin and eosin (H&E) staining. PT and DT depict the postal and distal tubules, luminal debris, and elevated tubular lumen, indicated by the blue and red arrow. (**B**) Dihydroethidium (DHE) and (**C**) acridine orange (AO) fluorescent staining. (**D**) Senescence-associated β-galactosidase (SA- β-gal) staining. Quantification of (**E**) DHE and (**F**) AO fluorescent intensities and (**G**) SA-β-gal-stained area. HC represents the high-cholesterol diet, HC + 1.0% RYR or 1.0% PCO represents the high-cholesterol diet supplemented with 1.0% red yeast rice or 1.0% policosanol, and HC + 0.5% RYR + 0.5% PCO represents the high-cholesterol diet supplemented with 0.5% each of red yeast rice and policosanol. Statistical significance at *p* < 0.05 (*) and *p* < 0.001 (***) compared to the HC group; ^†††^ (*p* < 0.001) represents the statistical significance compared to the 1.0% RYR group; ns represents a non-significant (*p* > 0.05) difference between the groups.

**Figure 9 pharmaceuticals-18-00200-f009:**
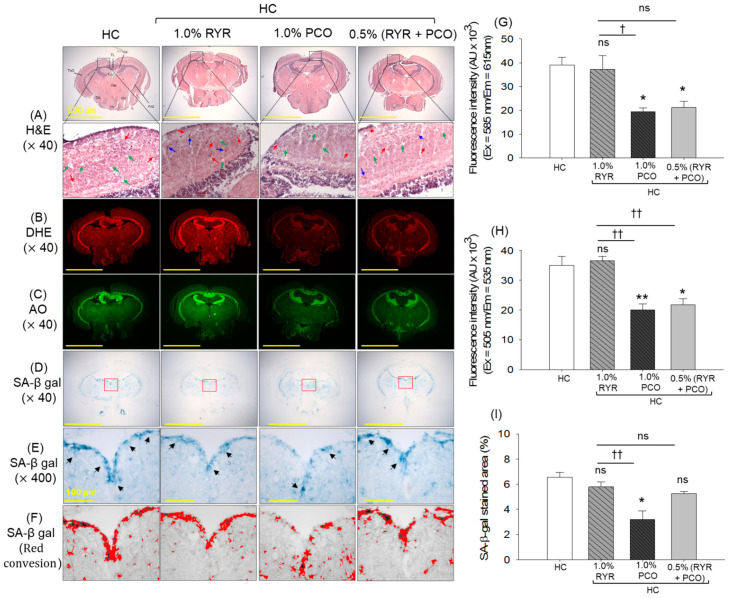
Brain histology of zebrafish following 12-week supplementation of red yeast rice (RYR) and policosanol (PCO) under a high-cholesterol diet. (**A**) Hematoxylin and eosin (H&E) staining. Blue arrow displayed an elevation in vacuolation, red arrow indicates number of mononuclear cells with a clear zone and the green arrow indicates the hyperemia. (**B**) Dihydroethidium (DHE) and (**C**) and acridine orange (AO) fluorescence-stained images. (**D**) Senescence-associated β-galactosidase (SA-β-gal)-stained area visualized at 40× magnification [500 μm, scale bar]. The red box in (**D**) is 400× magnified and images (**E**) represents this magnified view. (**E**) SA-β-gal-stained area visualized at 400× magnification [100 μm, scale bar]. (**F**) The SA-β-gal-stained blue color (at 400× magnification) interchanged with red color (at blue color threshold value 0–120) to intensify the visibility. Quantification of (**G**) DHE and (**H**) AO fluorescent intensities and (**I**) SA-β-gal-stained area. HC represents the high-cholesterol diet, HC + 1.0% RYR or 1.0% PCO represents the high-cholesterol diet supplemented with 1.0% red yeast rice or 1.0% policosanol, and HC + 0.5% RYR + 0.5% PCO represents the high-cholesterol diet supplemented with 0.5% each of red yeast rice and policosanol. Statistical significance at *p* < 0.05 (*), and *p* < 0.01 (**) compared to the HC group; ^†^ (*p* < 0.05), and ^††^ (*p* < 0.001) represents the statistical significance compared to the 1.0% RYR group; ns represents a non-significant (*p* > 0.05) difference between the groups.

**Figure 10 pharmaceuticals-18-00200-f010:**
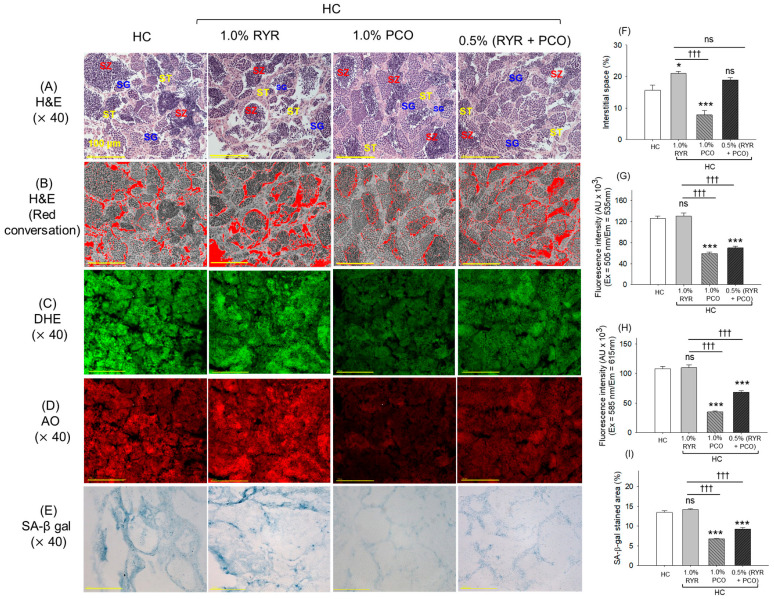
Testis histology of zebrafish after 12-week supplementation of red yeast rice (RYR) and policosanol (PCO) under the high-cholesterol diet. (**A**) Hematoxylin and eosin (H&E) staining. Spermatocytes, spermatogonia, and spermatozoa are abbreviated as ST, SG, and SZ, respectively. (**B**) The interstitial area was red-converted into the H&E section by using Image J software (at white color threshold of 220–255) to enhance visibility. (**C**) Dihydroethidium (DHE) and (**D**) acridine orange (AO) fluorescent staining. (**E**) Senescence-associated β-galactosidase (SA-β-gal) staining. (**F**) Quantification of interstitial space between the seminiferous tubules. Quantification of (**G**) DHE and (**H**) AO fluorescent intensities. (**I**) Quantification of the SA-β-gal-stained area. HC represents the high-cholesterol diet, HC + 1.0% RYR or 1.0% PCO represents the high-cholesterol diet supplemented with 1.0% red yeast rice or 1.0% policosanol, and HC + 0.5% RYR + 0.5% PCO represents the high-cholesterol diet supplemented with 0.5% each of red yeast rice and policosanol. Statistical significance at *p* < 0.05 (*) and *p* < 0.001 (***) compared to the HC group; ^†††^ (*p* < 0.001) represents the statistical significance compared to the 1.0% RYR group; ns represents a non-significant difference (*p* > 0.05) difference between the groups.

**Figure 11 pharmaceuticals-18-00200-f011:**
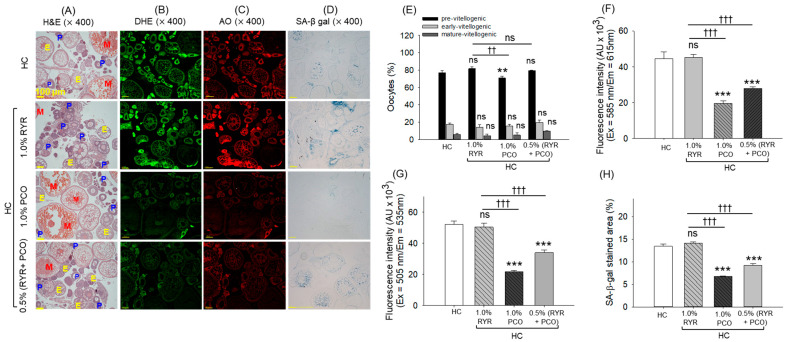
Ovary histology of adult zebrafish consuming red yeast rice (RYR) and policosanol (PCO) for 12 weeks under the influence of a high-cholesterol diet. (**A**) Hematoxylin and eosin (H&E) staining; pre-, early, and mature vitellogenic oocytes are represented the P, E, and M, respectively. (**B**) dihydroethidium (DHE) and (**C**) and acridine orange (AO) staining. (**D**) Senescence-associated β-galactosidase (SA-β-gal) staining [100 μm, scale bar]. (**E**) Oocyte count in the H&E-stained area. (**F**) Quantification of DHE and (**G**) AO fluorescent intensities. (**H**) Quantification of the SA-β-gal-stained area. HC represents the high-cholesterol diet, HC + 1.0% RYR or 1.0% PCO represents the high-cholesterol diet supplemented with 1.0% red yeast rice or 1.0% policosanol, and HC + 0.5% RYR + 0.5% PCO represents the high-cholesterol diet supplemented with 0.5% each of red yeast rice and policosanol. Statistical significance at *p* < 0.001 (**) and *p* < 0.001 (***) compared to the HC group; ^††^ (*p* < 0.001) and ^†††^ (*p* < 0.001) represent the statistical significance compared to the 1.0% RYR group; ns represents a non-significant difference (*p* > 0.05) between the groups.

## Data Availability

The data used to support the findings of this study are available from the corresponding author upon reasonable request.

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
