# Peer review of "A Comparative Effect of 12-Week Dietary Intervention of Policosanol (Raydel®) and Red Yeast Rice (RYR, Kobayashi) in Managing Dyslipidemia and Organ Damage in Hyperlipidemic Zebrafish"

_pharmaceuticals, 2025, doi:10.3390/ph18020200_

Round 1
Reviewer 1 Report (Previous Reviewer 1)
Comments and Suggestions for Authors
The revised version of the manuscript is greatly improved, first in the definition of rationale of some choices, and the authors’ explanations have clarified the issues that needed some highlighting.
References’ imprecisions have been amended, and some new added citations are useful for a better understanding of the trial.
Considering all the pending issues as amended, the manuscript can be recommended for publication
Reviewer 2 Report (Previous Reviewer 2)
Comments and Suggestions for Authors
The manuscript presents a thorough study comparing the effects of red yeast rice (RYR) and Cuban policosanol (PCO) on hyperlipidemic zebrafish, focusing on lipid regulation, antioxidant status, and organ health. The findings highlight the therapeutic efficacy of PCO, which improved lipid profiles, enhanced antioxidant markers, and provided hepatoprotective effects, while mitigating oxidative stress and histological damage across multiple organs. In contrast, the study identifies significant toxicological concerns with RYR, including hepatic and renal damage, underscoring the importance of safety assessments in dietary interventions. The integration of biochemical and histological analyses provides robust evidence supporting PCO as a safer and more effective option for managing hyperlipidemia. Overall, the manuscript makes a valuable contribution to the field of nutritional therapeutics and is well-suited for publication.
This manuscript is a resubmission of an earlier submission. The following is a list of the peer review reports and author responses from that submission.
Round 1
Reviewer 1 Report
Comments and Suggestions for Authors
The manuscript reports the results of a comparative trial with two lipid-lowering nutraceuticals, that is always topic. Nonetheless, as authors themselves state in the text, these molecules and their comparison have been performed many times and scientific literature is available (“policosanol has been extensively explored in preclinical and clinical studies”, “The safety aspect of policosanol is studied in wide animal models like rats, dogs, and monkeys” and “Owing to the several beneficial functions and nontoxic effects, policosanol is gaining acceptability as a nutraceutical globally, as evidenced by the approval of policosanol by 25 countries as a safe cholesterol-lowering agent”, together with “Due to the possible toxicity and health implication the United State of Food and Drug Administration (FDA) provided a specific guideline against the RYR supplementation for hyperlipidemia”,” The results are aligned with the reports showing the fatal effect of RYR consumption [33,34]. Particularly, in early 2024, the Kobayashi RYR scandal has been reported with 80 deaths and more than 500 people hospitalized following the Kobayashi Beni-Koji RYR supplement, posed a serious safety concern about RYR based supplements. Due to these adverse events Kobayashi pharmaceuticals stopped the production of Beni-Koji RYR from August 2024 and withdraw all the RYR product from the global market [35] and “Despite the varied studies (non-clinical and clinical) using either policosanol or RYR has been performed”), so what is the real utility of another comparative study? On the contrary, there is something weird in the citations, especially those on toxic effects of RYR on humans:
· Cit. 26 is a Review of the state-of-the-art on RYR (2024) and does not mention Coleste help by Kobayashi Co. ...
· L 93-95: “Despite this the uncertain toxicity or RYR consumption is always being a concern. Some clinical studies reported hepatotoxicity, respiratory injury, reproductive toxicity, musculoskeletal and gastrointestinal toxicity raising the serious concern of RYR consumption [28], but 28 reports “Acute kidney injury associated with red yeast rice (Beni-kōji) supplement: A report of two cases. Kidney Med. 2024, 6, 100908, Uchiyama, K.; Otani, M.; Chigusa, N.; Sugita, K.; Matsuoka, R.; Hosoya, K.; Komuta, M.; Washida, N”
· L 501-507: the cited reports on fatalities connected with the use of RYR (33 and 34) do not even mention Red Yeast Rice, while 35 is a short single case report where the authors outline only the similarity of monacolin K with lovastatin and presumably the share of the same hepatotoxicity. On the contrary, 36 is a brief article describing the case of numerous health hazard events connected with Kobayashi RYR products, but also reports “Japanese mass media assumed that these adverse events were attributable to puberulic acid, a compound produced by blue mould, thereby suggesting contamination at the production facility as a potential source”, so a casualty linked with the lack of regulated controls on the production of nutraceuticals and not with some intrinsic toxicity of RYR. Please, add some explanation and/or more correct citations. Moreover, was the batch used to supplement fish food analyzed for puberulic acid contamination/content? This assessment should have been considered as fundamental in a trial that aims to ascertain the toxicity of RYR, where this occasional contaminant could be the real responsible of the observed adverse effects
· Cit. 37 is a Wikipedia page, not scientific literature: https://en.wikipedia.org/wiki/Kobayashi_red_yeast_rice_scandal
Images and graphs are clear
Overall, the English language is fine and fluent, with some typo (e.g., see 15) or incorrect sentences (592-593, 607-608, 612-614)
References are updated, but not appropriate in their citation
Some specific comment:
15: what is the meaning for “hyperendemic”? please, explain
85-86: “nutraceutical/functional food with the hypercholesterolemic effects”? please, check the sentence
120-124: see comment to 657-660
625-626: considering that Kobayashi's RYR products have been implicated in toxicity/fatality incidents, likely related to production contamination, why do authors choose precisely this manufacturer’s RYR? Please, explain
641-646: Tetrabits is a granular fish food, how was it compounded with supplemental cholesterol and RYR/PCO? Please, add description of methods
643-644: how was the 1% w/w dose decided?
657-660: superficial mucus is a fundamental part of cutaneous defense system of fishes, is there an ethical approval for its damaging during the drying procedures for weight measurement? Furthermore, the surface drying with tissue paper cannot be uniform due to the partial removal of the mucus itself with the paper, so the evaluation of some mg variation in the weight of a small fish like zebra can be inaccurate
The general impression is of serious lack of precision in the premises and in the performance of the in vivo part, that can generate false positive results
Reviewer 2 Report
Comments and Suggestions for Authors
This study examines the impact of a 12-week dietary intervention with RYR and PCO on lipid profiles, antioxidant status, and organ health in hyperlipidemic zebrafish. Both supplements effectively reduced total cholesterol (TC), triglycerides (TG), and low-density lipoprotein cholesterol (LDL-C). Notably, PCO increased high-density lipoprotein cholesterol (HDL-C) and improved antioxidant markers. There are some major comments.
The lipid-lowering effects of RYR and PCO are well-documented. The study should clearly articulate the novel aspects of this research, particularly, the comparative analysis between RYR and PCO, or any unique findings regarding their mechanisms of action.
Clarification is needed on how the dosages for RYR and PCO were established. Specifically, do these dosages correspond to typical human consumption levels, and how were they scaled for zebrafish?
It would be beneficial to compare the efficacy of RYR and PCO with standard dyslipidemia treatments, such as statins.
In the methodology section, specify the name of the ethical review committee or institutional review board that approved the study, along with the approval number.
The study should discuss how the lipid profile changes and organ responses observed in zebrafish correlate with human dyslipidemia.
The term "hyperendemic zebrafish" used in the abstract is unconventional.
The study lacks an exploration of the molecular mechanisms underlying hyperlipidemia and the action of RYR and PCO.
Figure 3 indicates an increase in blood glucose levels in the HC + 0.5% RYR + 0.5% PCO group. The manuscript should address potential reasons for this observation, considering possible interactions between RYR and PCO or other metabolic effects.
In Figure 6, the hepatic reactive oxygen species (ROS) staining using DHE and AO appears to show increased brightness rather than specific fluorescence signals.
Comments on the Quality of English Language
Can be improved. Typo errors and Grammatical errors need to be checked throughout the manuscript